# V-MAO: Generative Modeling for Multi-Arm Manipulation of Articulated Objects

**Xingyu Liu**
Robotics Institute
Carnegie Mellon University
xingyul3@cs.cmu.edu

**Kris M. Kitani**
Robotics Institute
Carnegie Mellon University
kkitani@cs.cmu.edu

**Abstract:** Manipulating articulated objects requires multiple robot arms in general. It is challenging to enable multiple robot arms to collaboratively complete manipulation tasks on articulated objects. In this paper, we present V-MAO, a framework for learning multi-arm manipulation of articulated objects. Our framework includes a variational generative model that learns contact point distribution over object rigid parts for each robot arm. The training signal is obtained from interaction with the simulation environment which is enabled by planning and a novel formulation of object-centric control for articulated objects. We deploy our framework in a customized MuJoCo simulation environment and demonstrate that our framework achieves a high success rate on six different objects and two different robots. We also show that generative modeling can effectively learn the contact point distribution on articulated objects.

**Keywords:** Articulated object, generative model, variational inference

## 1 Introduction

In robotics, one of the core research problems is how to endow robots with the ability to manipulate objects of various geometry and kinematics. Compared to rigid objects, articulated objects contain multiple rigid parts that are kinematically linked via mechanical joints. Because of the rich functionality due to joint kinematics, articulated objects are used in many applications. Examples include opening doors, picking up objects with pliers, and cutting with scissors, etc.

While the manipulation of a known rigid object has been well studied in the literature, the manipulation of articulated objects still remains a challenging problem. Suppose an articulated object has $N$ rigid parts. In the most general case, it requires $N$ robot grippers to manipulate the $N$ object parts where each gripper grasps one rigid part to fully control all configurations of the articulated object, including position and joint angles [1]. How to teach the robot arms to collaborate safely without collision is still a research problem [1, 2]. Our intuition is that, instead of a deterministic model, sampling from a distribution of grasping actions can provide more options and can increase the chance of finding feasible collaborative manipulation after motion planning. Therefore, the generative modeling of grasping can be a solution to the problem.

We hypothesize that an object-centric representation of contact point distribution contains full information about possible grasps and can generalize better across different objects. Therefore it is preferred over robot-centric representation such as low-level torque actions, due to the uncertainty of robot configurations. Specifically for articulated objects, we further hypothesize that the grasping distribution on one rigid part is conditioned on the geometric and kinematics feature as well as grasping actions on other parts, and can be learned from interacting with the environment.

In this paper, we propose a framework named *V-MAO* for learning manipulation of articulated objects with multiple robot arms based on an object-centric latent generative model for learning grasping distribution. The latent generative model is formulated as a conditional variational encoder

---

[1]Though in certain scenarios, it is still possible that an articulated object be manipulated by a single gripper, e.g. dexterous manipulation of a pair of scissors with one human hand. However, such manipulation is extremely complex and is not scalable for other arbitrary articulated objects.

5th Conference on Robot Learning (CoRL 2021), London, UK.

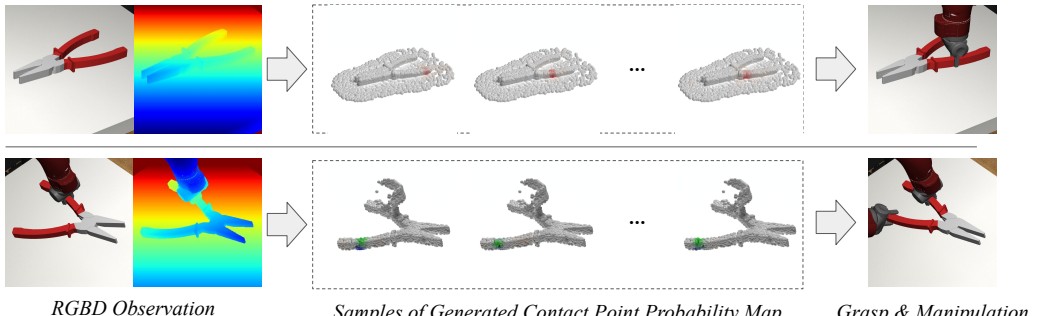

*RGBD Observation*        *Samples of Generated Contact Point Probability Map*        *Grasp & Manipulation*

Figure 1: **Generative Modeling of Multi-Arm Manipulation of Articulated Objects.** Given the RGBD scan observation of the articulated object and the scene, we use a generative model to learn the distribution of feasible contacts of each robot arm. The generative model for the second robot arm (second row) is conditioned on the action performed by the first robot arm (first row).

(CVAE), where the distribution of contact probability on the 3D point clouds of one object rigid part is modeled by variational inference. Note that the contact probability distribution on one object part is conditioned on the grasping actions already executed on other object parts which can be observed from the current state of 3D geometric and kinematics features. To obtain enough data for training, our framework enables automatic collection of training data using the exploration of contact points and interaction with the environment. To enable interaction, we propose a formulation of *Object-centric Control for Articuated Objects (OCAO)* to move the articulated parts to desired poses and joint angles after successful grasp. The framework is illustrated in Figure 1.

We demonstrate our V-MAO framework in a simulation environment constructed with MuJoCo physics engine [3]. We evaluate V-MAO on six articulated objects from PartNet dataset [4] and two different robots. V-MAO achieves over 80% success rate with Sawyer robot and over 70% success rate with Panda robot. Experiment results also show that the proposed variational generative model can effectively learn the distribution of successful contacts. The proposed model also shows advantage over the deterministic model baseline in terms of success rate and the ability to deal with variations of environments.

The contributions of this work are three-fold:

1. A latent generative model for learning manipulation contact. The model extracts articulated object geometric and kinematic representations based on 3D features. The model is implemented based on a conditional variational model.

2. A mechanism for automatic contact label generation from robot interaction with the environment. We also propose a formulation of Object-centric Control for Articulated Objects (OCAO) that enables the interaction by the controllable moving of articulated object parts.

3. We construct a customized MuJoCo simulation environment and demonstrate our framework on six articulated objects from PartNet and two different robots. The proposed model shows advantage over deterministic model baseline in terms of success rate and dealing with environment variations.

## 2 Related Works

**Generative Modeling of Robotic Manipulation.** Previous work has proposed to leverage latent generative models such as variational auto-encoder (VAE) [5] and generative adversarial network (GAN) [6] in various aspects of robotic manipulation. Wang et al. [7] proposed to use Causal InfoGAN [8] to learn to generate goal-directed object manipulation plan directly from raw images. Morrison et al. [9] proposed a generative convolutional neural net (CNN) for synthesizing grasp from depth images. Mousavian et al. [10] use VAE to model the distribution of pose of 6D robot grasp pose. Other works have proposed to use generative model to learn more complex grasping tasks such as multi-finger grasping [11] and dexterous grasping [12, 13]. Generative modeling can also be seen in other aspects of robotic manipulation such as for object pose estimation [14] and robot gripper design [15]. Our approach focuses on using variational latent generative model on 3D point clouds for learning object contact point distributions on articulated objects.

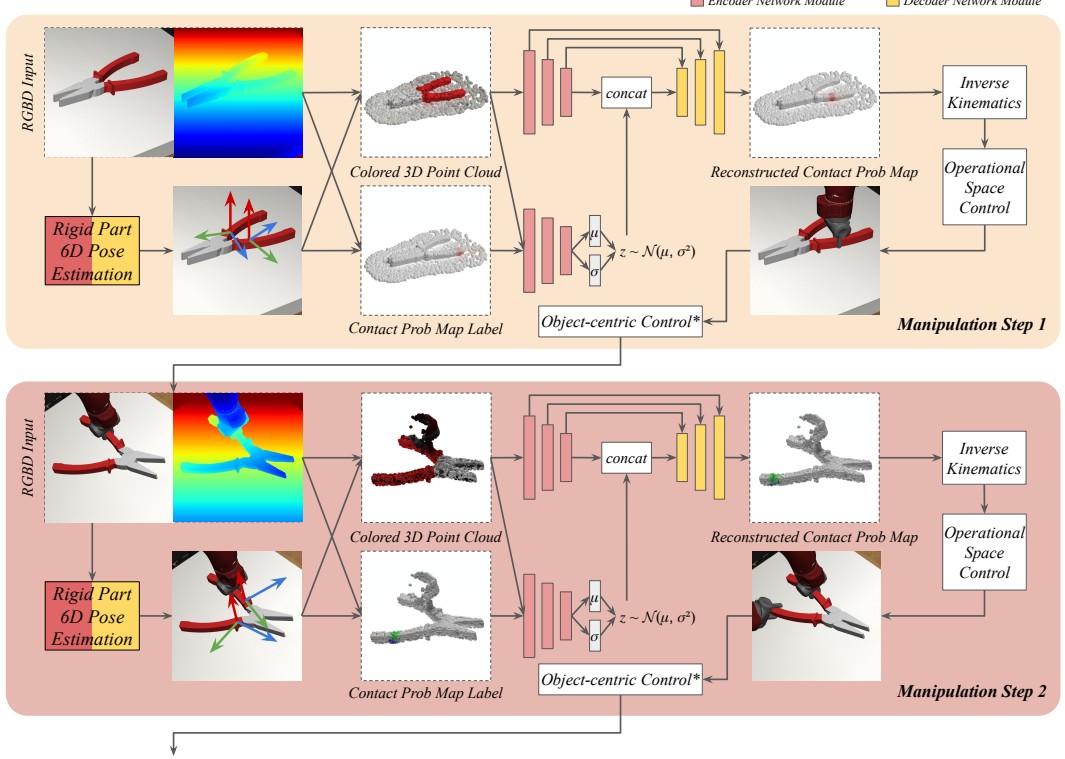

Figure 2: **Architecture of V-MAO Framework.** Given the scene point cloud, each step of the manipulation is modeled by a conditional VAE. Interaction with the object is done through inverse kinematics and object-centric control. Given the manipulation of one object part with a robot is successful, V-MAO will move to the manipulation of the next object part with the next robot.

**Articulated Object Manipulation.** The task of manipulating articulated object has been studied on both perception and action perspective in the literature. For example, Katz et al. [16] proposed a relational state representation of object articulation for learning manipulation skill. From interactive perception, Katz et al. [17] learns kinematic model of an unknown articulated object through interaction with the object. Kumar et al. [18] propose to estimate mass distribution of an articulated objects through manipulation. Narayanan et al. [19] introduced kinematic graph representation and an efficient planning algorithm for object articulation manipulation. More recently, Mittal et al. [20] constructed a mobile manipulation platform that integrates articulation perception and planning. Our work models contact point distributions from object geometry using a generative model and propose an object-centric control formulation for manipulating articulated objects.

**Multi-Arm in Robotic Manipulation.** Using multiple robot arms in manipulation tasks without collision is a challenging problem. Previous works have investigated using decentralized motion planner to avoid collision in multi-arm robot system [1]. Previous works have also investigated using multiple robots in the application of construction assembly [2], pick-and-place [21], and table-top rearrangement [22]. Our work is an application of using multiple robot arms to manipulate articulated objects where we focus on learning the contact point distribution on object parts.

## 3   Method

Our V-MAO framework consists of: 1) a variational generative model for learning contact distribution on each object rigd part (Section 3.2); and 2) inverse kinematics, planning, and control algorithms for interacting with the articulated object (Section 3.3). In Section 3.1, we first formally define the problem of using the generative model for articulated object manipulation. In Section 3.4, we provide additional details to our framework. The framework is illustrated in Figure 2.

### 3.1 Problem Definition

We consider the problem of multi-arm manipulation of articulated objects. The object has $N$ rigid parts and the system contains $N$ robot arms where $N \geq 2$. Robot arm $i$ executes high-level grasping action $g_i$ and moving actions $a_i$ on the objects. Grasping action $g_i$ is represented by the contact points on object part $i$. Moving actions $a_i$ send torque signals to robot $i$ to move object part $i$ to desired 6D pose $\mathcal{T}_i$. The whole manipulation is a sequence of $g_i$ and $a_i$.

Given colored 3D point clouds from RGBD sensors, suppose $\mathbf{p}_i = \{\mathbf{x}_{i,j} \in \mathbb{R}^{3+C}, j = 1, \ldots, n_i\}$ is the 3D point cloud of the $i$-th rigid part of the object, where $n_i$ is the number of points and $C$ is the number of feature channels, e.g. color values and motion vectors. Suppose $\mathbf{p}_S^{(k)} = \{\mathbf{x}_j^{(k)} \in \mathbb{R}^{3+C}\}$ is the 3D point cloud of the non-object scene and $\mathcal{T}_i^{(k)}$ is the 6D pose of $i$-th rigid object part after executing grasping actions $g_1, g_2, \ldots, g_k$. Note that the complete 3D point cloud of the scene $\mathbf{p}^{(k)} = \mathbf{p}_S^{(k)} \bigcup (\bigcup_{i=1} \mathbf{p}_i) = \mathcal{O}(\prod_{l=1}^{k} g_l, \prod_{l=1}^{N} \mathcal{T}_l^{(k)})$ is a function of grasp actions $(g_1, g_2, \ldots, g_k)$ and object part poses $\mathcal{T}_i^{(k)}$, where $\mathcal{O}$ is the observation of the environment and depends on robot forward kinematics. We assume there is no cycle in object joint connections so that there is no cyclic dependency among object parts. We formulate the manipulation problem as learning the following joint distribution of $(g_1, g_2, \ldots, g_n)$ given initial scene point cloud $\mathbf{p}^{(0)}$:

$$P(g_1, \ldots, g_n \mid \mathbf{p}^{(0)}) = P_1(g_1 \mid \mathbf{p}^{(0)}) P_2(g_2 \mid \mathbf{p}^{(1)}) \cdots P_n(g_n \mid \mathbf{p}^{(N-1)}) \tag{1}$$

$P_i$ is the contact point distribution model of $i$-th rigid part. In Equation (1), we use conditional probability to decompose $P$ and assume previous point clouds have no effect on the current grasping distribution. Our goal is to learn distribution $P_i$ for each object part that covers the feasible grasping as widely as possible so that sampled grasping actions $g_i$ and subsequent moving actions $a_i$ can successfully take all object parts to desired poses $\mathcal{T}_i$.

### 3.2 Generative Modeling of Contact Probability

Our generative model is based on variational inference [5]. Different from 2D images where the positions of the pixels are fixed and only the pixel value distribution needs to be modeled, in 3D point clouds, both geometry and point feature distribution need to be modeled. Therefore, instead of directly using multi-layer perceptions (MLPs) on pixel values similar to vanilla VAE, our model needs to learn both local and global point features in hierarchical fashion [23]. Moreover, in our model, the reconstructed feature map should not only be conditioned on latent code, but hierarchical geometric features as well.

**Encoder and Decoder for One Single Step.** Given a colored 3D point cloud appended with successful grasping probability maps $\mathbf{p} = \{\mathbf{x}_j \in \mathbb{R}^{3+4}\}$ where four feature channels are RGB and probability values, our model split the feature channels to construct two point clouds appended with different features: a 3D color map $\mathbf{p}_{rgb} = \{\mathbf{x}_j \in \mathbb{R}^{3+3}\}$ as the input to geometric learning, and $\mathbf{p}_c = \{\mathbf{x}_j \in \mathbb{R}^{3+1}\}$ as the 3D contact probability map to reconstruct using generative model.

An encoder network $\mathcal{F}_c(\cdot; \theta_c)$ is used to approximate the posterior distribution $q(z \mid \mathbf{p}_c, \mathbf{p}_{rgb})$ of the latent code $z$. Another encoder network $\mathcal{F}_{rgb}(\cdot; \theta_{rgb})$ learns the hierarchical geometric representation and outputs features $\mathbf{h}_1, \mathbf{h}_2, \mathbf{h}_3$ at different levels:

$$[\mu, \sigma] = \mathcal{F}_c(\mathbf{p}_c, \mathbf{p}_{rgb}; \theta_c), \ [\mathbf{h}_1, \mathbf{h}_2, \mathbf{h}_3] = \mathcal{F}_{rgb}(\mathbf{p}_{rgb}; \theta_{rgb}) \tag{2}$$

where $\mu$ and $\sigma$ are the predicted mean and standard deviation of the multivariate Gaussian distribution respectively. The latent code $z$ is then sampled from the distribution and is used to reconstruct the contact probability map on point clouds with a decoder network $\mathcal{G}(\cdot; \theta_g)$:

$$z \sim \mathcal{N}(\mu, \sigma^2), \ \hat{\mathbf{p}}_c = \mathcal{G}(z, \mathbf{h}_1, \mathbf{h}_2, \mathbf{h}_3; \theta_g) \tag{3}$$

Note that both $\mathcal{F}_c(\cdot; \theta_c)$ and $\mathcal{G}(\cdot; \theta_g)$ are conditioned on $\mathbf{p}_{rgb}$. Therefore, our generative model can be viewed as a conditional variational encoder (CVAE) [24] that learns the latent distribution of the 3D contact probability map $\mathbf{p}_c$ conditioned on the deep features of 3D color map $\mathbf{p}_{rgb}$. The conditional variational formulation takes observation $\mathbf{p}_{rgb}$ into account and can improve the generalization.

The encoders $\mathcal{F}_{rgb}(\cdot; \theta_{rgb})$ and $\mathcal{F}_c(\cdot; \theta_c)$ share the same architecture and are instantiated with set abstraction layers from PointNet++ [23]. The Decoder $\mathcal{G}(\cdot; \theta_g)$ is instantiated with set upconv layers proposed in [25] to propagate features to existing up-sampled point locations in a learnable fashion.

**Loss Function.** Our generative model is trained to maximize the log-likelihood of the generated contact probability map to be successful contact labels. Due to Jensen's inequality, the evidence lower bound objective (ELBO) in variational inference can be derived as

$$\log P(\hat{\mathbf{p}}_c) \geq E_{z \sim q(z|\mathbf{p}_c, \mathbf{p}_{\text{rgb}})} \log P(\hat{\mathbf{p}}_c|z, \mathbf{p}_{\text{rgb}}) - D_{KL}(q(z|\mathbf{p}_c, \mathbf{p}_{\text{rgb}})||\mathcal{N}(0,1)) = -\mathcal{L} \quad (4)$$

The total loss $\mathcal{L} = \mathcal{L}_{\text{recon}} + \mathcal{L}_{\text{latent}}$ consists of two parts. Reconstruction loss $\mathcal{L}_{\text{recon}} = -E_{z \sim q(z|\mathbf{p}_c, \mathbf{p}_{\text{rgb}})} \log P(\hat{\mathbf{p}}_c|z, \mathbf{p}_{\text{rgb}}) = -E_{z \sim q(z|\mathbf{p}_c)} \log P(\hat{\mathbf{p}}_c|z) = H(\mathbf{p}_c, \hat{\mathbf{p}}_c)$ is the cross entropy between $\mathbf{p}_c$ and $\hat{\mathbf{p}}_c$. Latent loss $\mathcal{L}_{\text{latent}} = D_{KL}(\mathcal{N}(\mu, \sigma^2)||\mathcal{N}(0,1))$ is the KL Divergence of two normal distributions. The goal of the training is to minimize the total loss $\mathcal{L}$. The parameters in the three networks, $\theta_{\text{rgb}}$, $\theta_{\text{c}}$ and $\theta_{\text{g}}$, are trained end-to-end to minimize $\mathcal{L}$.

### 3.3 Planning and Control Actions

Given the predicted contact points on object part $i$, the 6D pose of the end-effector of robot $i$ can be obtained by solving an inverse kinematics problem. Path planning algorithm such as RRT [26] is used to find an operational space trajectory from initial pose to desired grasping pose. If a path is not found, the framework will iteratively sample contact point distributions from the generative model multiple times. Then we use Operational Space Controller [27] to move the gripper along the trajectory and finally complete the grasping action.

After a successful grasp, the next goal is to apply torques on robot arms and gripper to move the object part $i$ to the desired pose. To keep the grasping always valid, there should be no relative motion between the gripper and the object part. We propose **Object-centric Control for Articulated Objects (OCAO)** formulation. It computes the desired change of the 6D poses of robot end effectors given the desired change of the 6D pose of one of the object rigid parts and the changes of all joint angles. Operational Space Control (OSC) [27] will then be used to execute the desired pose change of the robot end effectors. For more technical details on the mathematical formulation of OCAO, please refer to the supplementary material.

### 3.4 Additional Implementation Details

**Part-level Pose Estimation.** To ensure the object contact point is present in the 3D point cloud, we merge the colored object mesh point clouds and the scene point cloud as a combined point cloud. The 6D pose of each object part is estimated using DenseFusion [28], an RGBD-based method for 6D object pose estimation. DenseFusion is trained separately using the simulation synthetic data. Since adding object mesh point cloud to the scene point cloud causes unnatural point density, we use farthest point sampling (FPS) on the merged scene point cloud to ensure uniform point density.

**Automatic Label Generation from Exploration.** Training the variational generative model requires a large amount of data. To collect sufficient data for training, we propose to automatically generate labels of contact point combinations using random exploration and interaction with the simulation environment. Expert-defined affordance areas of grasping, for example the handle of the plier, are provided to ensure realistic grasping and narrow down the exploration range. We provide initial human-labeled demonstrations of possibly feasible contact point combinations, for example a pair of points at the opposite sides of the plier handle, to speed up exploration efficiency. We then use random exploration on the contact points within the affordance region and verify the contact point combinations through interaction in the simulation environment. Successful contact point combinations are saved in a pool used in training. For more details on the algorithm of random exploration, please refer to the supplementary material.

## 4 Experiments

We design our experiments to investigate the following hypotheses: 1) given training samples of enough variability, the contact point generative model can effectively learn the underlying distribution; 2) contact point labels can be automatically generated through interaction with the environment; 3) given sampled and generated contact points, after path planning and control, the robot is able to complete the manipulation task.

In this section, we present the experiments on manipulating *two-part* articulated objects from Part-Net [4] with *two* robot arms. To show that our framework can be applied to arbitrary number of

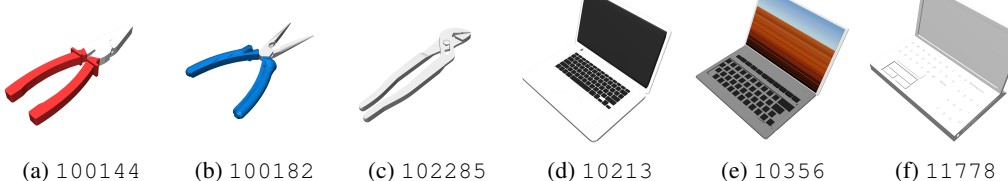

(a) `100144`  (b) `100182`  (c) `102285`  (d) `10213`  (e) `10356`  (f) `11778`

Figure 3: **The six articulated objects from PartNet [4] used in our experiments.** We illustrate their IDs in the dataset. The objects have two articulated parts connected by a revolute joint. The expert-defined affordance area is the handles for the three pliers and the non-keyboard base area for the three laptops.

articulated object parts and robot arms, we further conduct experiments on manipulating a *three-part* articulated object with *three* robot arms. Please refer to the supplementary material for the details on the three-arm experiments.

## 4.1 Experiment Setup

**Task and Environment.** We conduct the experiments in an environment constructed with MuJoCo [3] simulation engine. The environment consists of two robots facing a table in parallel. The articulated object is placed on the table surface with random initial positions and joint angles. The goal of the manipulation is to grasp the object parts and move them to reach the desired pose and joint angle configuration within a certain error threshold.

**Objects and Robots.** We evaluate our approach on six daily articulated objects from PartNet [4] in the experiments. The objects include three pliers and three laptops illustrated in Figure 3. For the robots in the experiment, we use Sawyer and Panda which are equipped with two-finger parallel-jaw Rethink and Panda grippers. Both robots and grippers are instantiated by high-fidelity models provided in robosuite [29]. For the friction between robot gripper and the object, we assume a friction coefficient of 0.65 and assume pyramidal friction cone.

**Baseline.** There is no related previous work on multi-arm articulated objects that we can compare our V-MAO with. So we developed a baseline approach denoted as "Top-1 Point" that predicts a deterministic per-point contact probability map on the point cloud using an encoder-decoder architecture and selects the best contact point for each gripper finger within the affordance region. To fairly compare, the baseline uses the same encoder and decoder architectures and is trained on the same interaction data as V-MAO. After contact point selection, the subsequent planning and control steps are also the same for the baseline.

## 4.2 Quantitative Evaluation

We explore two types of grasping strategy, sequential grasp and parallel grasp. In **sequential grasp**, object parts are grasped and moved in an sequential fashion. In $i$-th step, robot gripper $i$ grasps the $i$-th object part while leaving parts $i + 1$ through $N$ uncontrolled. We iterate the above step until all parts are grasped. In **parallel grasp** all object parts are grasped simultaneously. Then the robots move the object parts to the goal locations together.

We report the success rate of manipulation in Table 1. Our V-MAO can achieve more than 80% success rate with Sawyer robot and more than 70% success rate with Panda robot on all objects, which shows V-MAO can effectively learn the contact distribution and complete the task. Besides, on the Panda robot, sequential grasping generally achieves a much better success rate than parallel grasping. This means determining grasping for other robots later may be a better option than determining both robots' grasping at the beginning in certain cases.

We notice that when using the Panda robot, the performance of both V-MAO and the baseline decrease significantly. The reason is that the Panda robot has a large end effector size. Therefore when manipulating small objects like pliers, in many cases the space is not enough to contain both grippers due to robot collision, while Sawyer robot's end effector is much smaller. The deterministic prediction baseline achieves a larger success rate on laptops. A possible explanation is that there

| robot | method | grasp order | 100144 | 100182 | 102285 | 10213 | 10356 | 11778 |
|-------|--------|-------------|--------|--------|--------|-------|-------|-------|
| Sawyer | Top-1 Point | parallel | 90.0 | **92.0** | 86.0 | **96.0** | 88.0 | 90.0 |
| | | sequential | 82.0 | 78.0 | 88.0 | 92.0 | 84.0 | 86.0 |
| | V-MAO (Ours) | parallel | **94.0** | 90.0 | 90.0 | **96.0** | **90.0** | **94.0** |
| | | sequential | 92.0 | 86.0 | **92.0** | **96.0** | 88.0 | 90.0 |
| Panda | Top-1 Point | parallel | 50.0 | 20.0 | 48.0 | **90.0** | **92.0** | **88.0** |
| | | sequential | 62.0 | 42.0 | 70.0 | 78.0 | 82.0 | 70.0 |
| | V-MAO (Ours) | parallel | 52.0 | 36.0 | 64.0 | **90.0** | 84.0 | **88.0** |
| | | sequential | **72.0** | **76.0** | **78.0** | 88.0 | 86.0 | 86.0 |

Table 1: **Success rate of Manipulation.** The baseline we compare V-MAO with is selecting the point with the maximum contact probability from deterministic prediction. We report the results averaged from 50 runs.

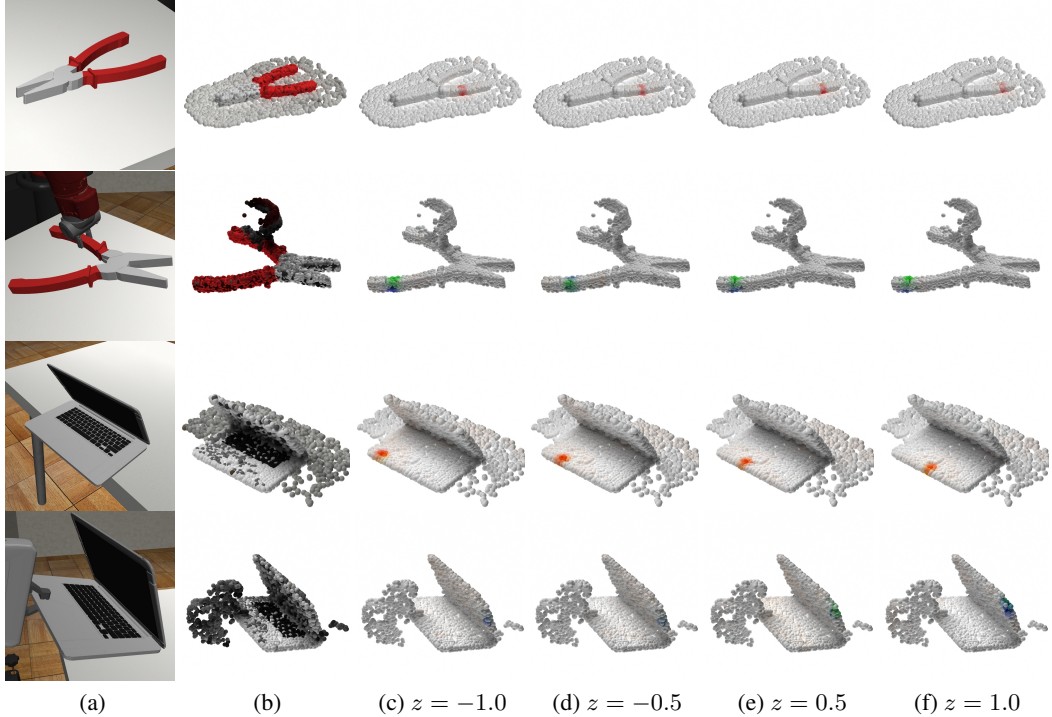

|        |        | (c) $z = -1.0$ | (d) $z = -0.5$ | (e) $z = 0.5$ | (f) $z = 1.0$ |
|--------|--------|----------------|----------------|---------------|---------------|
| (a) | (b) | | | | |

Figure 4: **Visualization of sampled and generated 3D probability maps.** From left to right in each row: (a) the current manipulation state; (b) converted colored 3D point cloud at the current state; (c)-(f) generated probability map of the next grasp given specific latent code $z$, where red and yellow denote the finger 1 and 2 of the first gripper in the first grasping step, and green and blue denote the finger 1 and 2 of the second gripper in the second grasping step. Zoom in for better a view of the figures.

are less variation in grasping laptops compared to pliers, for example the first grasp of the laptop is almost always grasping the base horizontally. Thus a deterministic model can be more stable in simpler tasks. On the other hand, in environments with larger variability, a generative model can have advantage.

### 4.3 Qualitative Results

In Figure 4, we visualize the probability map on the 3D point clouds sampled from the generative model given different latent codes $z$. As expected, the maximum value probability value falls on the object points. The points that have the highest probability for the respective gripper also form valid grasps. Given different latent codes $z$, the generated contact points can cover a wide range of feasible contact point combinations. We also notice that there are some object regions that are

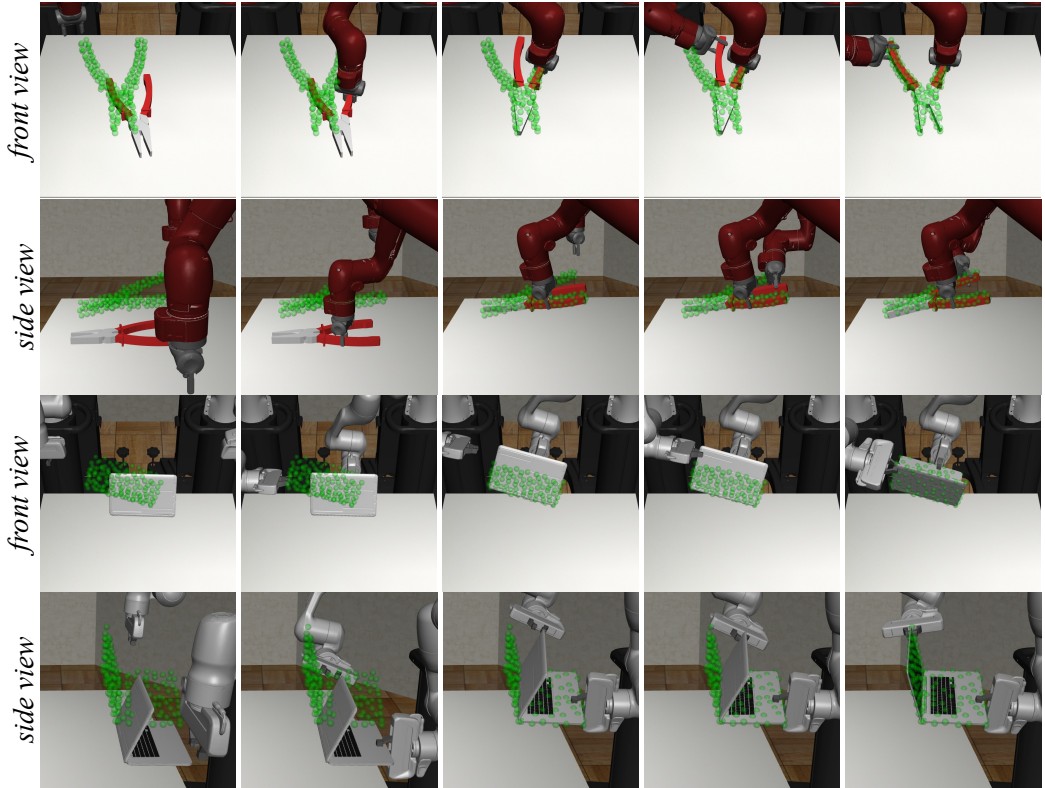

Figure 5: **Visualization of two sequential manipulations using two robot arms.** Rows 1 and 2 illustrate the manipulation of a pair of pliers (`100144`) using two Sawyer robots. Row 3 and 4 illustrate the manipulation of a laptop computer (`10213`) using two Panda robots. The object configuration goal of manipulation is marked by the green semi-transparent point cloud.

less covered by the generative model during sampling. One possible reason is that grasping in these regions can introduce collisions in the current or the next steps and have a lower probability of being sampled.

In Figure 5, we visualize the executed sequential manipulation using the contact points sampled from the generative model. As we can see, the robot gripper can successfully reach the contact points after inverse kinematics and path planning. When both grippers are grasping the object, the sampled and generated contact points combination for both robots can also avoid collision between robots. After grasping, both robots can reliably control the object to reach the goal configurations marked by green point cloud.

## 5   Conclusion

In this paper, we propose V-MAO, a framework for learning multi-arm manipulation of articulated objects. Our framework includes a variational generative model that learns contact point distribution over object rigid parts for each robot arm. We developed a mechanism for automatic contact label generation from robot interaction with the environment. We deploy our framework in a customized MuJoCo simulation environment and demonstrate that our framework achieves high success rate on six different objects and two different robots. V-MAO achieves over 80% success rate with Sawyer robot and over 70% success rate with Panda robot. Experiment results also show that the proposed variational generative model can effectively learn the distribution of successful contacts. As future work, we will investigate articulated object manipulation modeling using other sensor modalities such as stereo RGB images [30, 31] and dynamic point clouds [25, 32], or other pose estimation methods such as [33, 34].

**Acknowledgments**

This work is funded in part by JST AIP Acceleration, Grant Number JPMJCR20U1, Japan.

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
