# OpenReview forum: "V-MAO: Generative Modeling for Multi-Arm Manipulation of Articulated Objects"
_robot-learning.org/CoRL/2021/Conference — CoRL2021 Poster_

### Official Review · Reviewer_Fyb6 · 2021-07-23

**Originality:** Fair
**Technical Quality:** Very Good
**Clarity Of Presentation:** Fair
**Impact:** 3

**Recommendation:**

Weak Reject: I recommend rejecting the paper, but will not argue for my recommendation if the majority of other reviewers have a different opinion.

**Summary:**

This paper presents a generative approach to learn manipulation of articulated objects for multiple robot arms. The generative model learns a distribution of contact points on 3D point clouds. The approach is validated in simulated Mujoco environment.

**Issues:**

- Index of gk or gn in the problem formulation
- In Sec 3.3, I am confused by “another Operational Space Controller”. Does it mean the controller has a different parameter from the one for grasping? Or are they the same controller?
- The data collection is briefly mentioned, but I miss the details of how many training samples required for each object. Also, how the point clouds of the first robot arm is processed during data collection?
- I am not clear about the part of superimposing the object mesh point clouds. What does this step exactly mean?
- Some grammar issues such as, line 81 ‘object geometric’,  line 208 “iterative the above step”


**Reviewer Expertise:**

Very good: Comprehensive knowledge of the area

**Strengths And Weaknesses:**

Strengths:
- The approach seems very effective for multi-arm manipulation on articulated objects under two different grasping conditions.

Weakness:
- Section 2 shows a list of existing works on relevant topics, but it is confusing about the novelty of the proposed approach from the writing. Is the novelty on the side of learning a generative model on articulated objects? Or a generative model for grasping points? My understanding is that the contribution lies in the former one. In that case, what is the relationship between the proposed approach and some related works which also use generative approach on articulated objects (such as [1]) ?
- Before authors can address the concern between the current approach and existing works more extensively, the work seems only incremental to me, especially only evaluated in the simulation instead of on a real robot.
- Also, the formulation states that this is for multi-arm case where N>=2. However, all experiments are only done on two robot arms, without any quantitative or qualitative experiments to show three-arm cases. In that case, I think stating N>=2 is not convincing that the proposed approach can extend to N=3 case. I suggest authors either add more experiments or make more upfront formulation.


**Summary Of Recommendation:**

I rate clarity of presentation fair because at the current stage, it's hard to evaluate if the contribution is enough of the work. It is more like an incremental work, combining some generative approach and apply it in a bi-manual case. Overall the execution of the paper is good, but it fails to make the contribution clearly and also have some misleading statements such as the approach applies to N>=2 cases where there are no experiments to demonstration N>2 cases.

---

### Official Review · Reviewer_E39s · 2021-07-23

**Originality:** Very Good
**Technical Quality:** Very Good
**Clarity Of Presentation:** Very Good
**Impact:** 4

**Recommendation:**

Strong Accept: I recommend accepting the paper and will argue for my recommendation even if other reviewers hold a different opinion.

**Summary:**

This paper presents V-MAO, an architecture to learn multi-arm manipulation of articulated objects. It computes feature representations for 3D point clouds and a probability distribution over a latent variable, representing the contact points. Inverse Kinematics is used to compute the 6D pose of the robot end-effector given the predicted contact points on the object. RRT is used to compute collision-free paths. An Operational Space Controller is used to move the gripper. The 3D point cloud might contain only an object, with multiple parts, or also points corresponding to one of the grippers, which is important to achieve collision-free multi-arm manipulation.

The training data is obtained by using expert-defined affordance areas for grasping plus exploration by the robot.

The learning architecture is trained to maximize the log-likelihood of the generated contact points to lead to successful grasps.

**Issues:**

Can the authors discuss the main challenges to apply the proposed method to real robots?

Sometimes, the index "i" is used to denote several elements at the same time, e.g. robot i, object part i. Consider using different indexes to denote different elements.

In Section 3.1, why is P_S^(k) introduced. It does not seem to be used anywhere in the paper.

Is the latent variable just a scalar? Why not a vector? Wouldn't it be better to model the distribution over z by a multivariate Gaussian with full covariance matrix?

Can the authors make the subsection "Part-level Pose Estimation" more clear? What's the issue with superimposing object mesh points? How does farthest point sampling (FPS) work?

A visualization of the baseline architecture "Top-1 Point" in the paper or in the supplemental material would be helpful.

Section 4.2: "Besides, sequential grasping generally achieves a better success rate than parallel grasping. This means determining grasping for other robots later may be a better option than determining both robots’ grasping at the beginning." --> Actually, for the Sawyer robot, parallel was better (Table 1).

Typos:
- Abstract: " It is challenging to enable multiple robot arms to c collaboratively complete manipulation tasks on articulated objects" --> collaboratively
- 1. Introduction: "In this paper, we propose a framework name V-MAO for learning manipulation of articulated objects" --> named
- Figure 1: "Grasp & Manipuation" --> Grasp & Manipulation
- Section 2: "Other works has proposed to use generative model to learn more complex grasping" --> have proposed
- Section 2, Articulated Object Manipulation: "Katz et al. [15] proposed a relational state representation of of object articulation for learning manipulation skill" --> "of" is repeated
- Section 2, Articulated Object Manipulation: "For example, from interactive perception, Katz et al. [16] learns kinematic model of an unknown articulated object through interactive perception." --> "interactive perception" twice
- Section 2, Multi-Arm in Robotic Manipulation: "we focus on the learning the contact point distribution on object parts" --> focus on learning
- Section 3.1: "we use conditional probability to decompose P and assumes previous point clouds have no effect on the current grasping
distribution" --> assume
- Section 3.2: "only the pixel value distribution need to be modeled" --> needs
- Section 4: "We design our experiments to investigate the following hypothesis" --> hypotheses (plural)
- Figure 3: " We illustrates their ID in the dataset" --> illustrate
- Section 4.2: "We iterative the above step until all parts are grasped" --> iterate over the above step...
- Table 1: "Saywer" --> Sawyer

**Reviewer Expertise:**

Good: General knowledge of the area

**Strengths And Weaknesses:**

Strengths:
- This work tackles a challenging manipulation problem and produces convincing results in simulation.

Weaknesses:
- The presentation could be made more clear.
- The challenges on the way of applying this  method to real robots should be discussed.

Idea for future work:
- Consider multimodal distributions for the latent variable to be able to grasp at multiple disjoint positions.

**Summary Of Recommendation:**

The paper tackles a challenging problem in a technically sound manner and achieves very good results, as shown in the submitted video.

---

> ### Comment · Reviewer_E39s · 2021-09-01
> **Reviewer Response**
>
> Thank you for the additional info and experiments. My score remains the same.

---

### Official Review · Reviewer_War1 · 2021-07-25

**Originality:** Good
**Technical Quality:** Good
**Clarity Of Presentation:** Good
**Impact:** 4

**Recommendation:**

Weak Accept: I recommend accepting the paper, but will not argue for my recommendation if the majority of other reviewers have a different opinion.

**Summary:**

This paper aims to tackle the problem of multi-arm manipulation of articulated objects. The authors propose V-MAO, a framework that learns a variational generative model to predict the contact point distribution over object rigid parts for each robot arm to manipulate. They obtain the training signals via interactions with the simulation environment enabled by object-centric planning using the current model.

The authors deploy the framework in a customized MuJoCo simulation environment and demonstrate that it achieves a relatively high success rate on six different articulated objects with two rigid parts using two different robots. They have also shown that the generative model used in the framework allows it to outperform a deterministic baseline.

**Issues:**

I would love to see results on the real robots, but I understand that real-world deployment may be challenging due to the pandemic. Still, I hope the authors can address the issues raised in the main review, after which I would be willing to increase my score.

**Reviewer Expertise:**

Very good: Comprehensive knowledge of the area

**Strengths And Weaknesses:**

[Strength]

This paper targets an interesting yet challenging problem of manipulating articulated objects. The results in the simulation suggest that the model can manipulate several articulated objects and successfully achieve the desired configuration.


[Weakness]

While I like the direction this paper is going, I have concerns regarding its ability to deploy to the real world, the claims made in the paper, and the detailed design of the method.

This paper only shows experiments in simulation. Will the method generalize to the real world? Are there any missing components or potential problems when extending to the physical robots? The authors say that "the training signal is obtained from interaction with the simulation environment." Can the model trained in simulation directly generalize to the real world, or do we have to obtain the label via real-world interactions? The authors may consider providing more details on how much and what quality of interaction data is needed.

Related to my previous point, the method seems to require labels on object parts and the grasp probability map. Obtaining these labels in simulation may be straightforward but not in the real world. Can models trained using these labels in simulation generalize to the real world, or do we have to manually label them directly on the real data, and how well can the learned model generalize?

The authors claim that "V-MAO achieves over 80% success rate with Sawyer robot and over 70% success rate with Panda robot," and they have provided some videos of the successful trials, which is profoundly helpful. However, no failure examples are provided, making it hard to fully understand the performance of the method. The authors may consider showing the failure cases, analyzing them, and summarizing the common failure scenarios.

Related to my previous point, the authors may also consider providing failure cases for the deterministic baseline and include more explanations of why it is worse than V-MAO. The generative model is learned by giving the deterministic labels. Are there any additional supervisions or mechanisms that allow the proposed generative model outperforms the deterministic baselines?

The authors claim multiple times in the paper that the method is developed to handle objects with N rigid parts, where N >= 2. However, the authors only provide experimental results on objects consist of two rigid parts. Without concrete experimental results on objects with N > 2, I suggest the authors adjust their claims to set the right expectation for the readers.

The authors describe in Section 3.4 that the grasping areas are hand-specified, and they also need to provide initial demonstrations of feasible contact point combinations. These are important design choices for making the pipeline work. The authors may consider providing more details like how many demonstrations are needed for each object? What if there are multiple locations that could lead to successful grasps? Can it generalize to objects with similar but different geometries, e.g., objects in the same category?

How did the authors plan the grasps when given the grasping locations? Is it based on the local geometry of the object or expert demonstrations?

This paper contains an extensive amount of typos and grammatical errors. Detailed proofread would be necessary. Just to name a few:
Line 2: c collaboratively --> collaboratively;
Line 31: uncertainly --> uncertainty;
Line 42: desire poses --> desired poses;
Figure 2 caption: Each --> , each;
Line 82: a object-centric --> an object-centric;
Line 217: V-Mao --> V-MAO

============================

Post-rebuttal

I have read the reviews from other reviewers and the rebuttal from the authors. I appreciate the authors' efforts in providing the additional experimental results and the revision to the paper.

The authors' response has addressed most of my concerns, and I'm now supportive of this paper to present at the conference. I have changed my score from Weak Reject to Weak Accept.

**Summary Of Recommendation:**

While I like the direction this paper is going, I have concerns regarding its ability to deploy to the real world, the claims made in the paper, and the detailed design of the method. I hope the authors can address my issues, and I'm willing to increase the score to reflect the changes.

---

### Meta-Review · Area_Chair_GoQq · 2021-08-13

**Recommendation:** Accept (Poster)
**Confidence:** 4

**Metareview:**

Strengths:

- All of the reviewers felt the general direction of the work was interesting. There was a sense that the variational approach was interesting here.

- The reviewers felt that the simulated experiments effectively demonstrated that the approach works well in simulation.

Weaknesses:

- A key concern from the reviewers was that the approach was evaluated primarily in simulation. While there are real robot videos, there are little/no quantitative results.

- The AC had a question about the high level motivation of this paper. The approach assumes we are given full geometric object models. And, the method requires accurate part-level pose estimation (section 3.4). So, why not plan grasp points using traditional methods instead of the variational approach?

- The variational approach here seems closely related to the approach in “6-DOF GraspNet: Variational Grasp Generation for Object Manipulation” by Arsalan Mousavian, Clemens Eppner, Dieter Fox. However, in that work, the authors augment the variational sampling step w/ subsequent classification and grasp adjustment in order to get accurate grasp placement. Why was that not also necessary here?

- There was a question about where the ground truth labels came from for training the model. Were these hand-labeled? How large was the dataset / how many human demonstrations were provided?

- The baseline method, “Top-1 Point”, was not described very clearly in the main text and there is no other baseline.

- There was a concern that since the method is only evaluated for two robotic arms, it is unfair to assert the method works for larger numbers of arms.

Post-rebuttal:

We appreciate the new three-arm experiment strengthens the paper and the revisions for clarity.

---

> ### Comment · Reviewer_E39s · 2021-08-28
> **Are there real robot videos?**
>
> I only saw a video with simulation results. Are there any videos with results involving real robots?

---

### Decision · Program_Chairs · 2021-09-13

**Decision:**

Accept (Poster)

**Comment:**

Strengths:

- All of the reviewers felt the general direction of the work was interesting. There was a sense that the variational approach was interesting here.

- The reviewers felt that the simulated experiments effectively demonstrated that the approach works well in simulation.

Weaknesses:

- A key concern from the reviewers was that the approach was evaluated primarily in simulation. While there are real robot videos, there are little/no quantitative results.

- The AC had a question about the high level motivation of this paper. The approach assumes we are given full geometric object models. And, the method requires accurate part-level pose estimation (section 3.4). So, why not plan grasp points using traditional methods instead of the variational approach?

- The variational approach here seems closely related to the approach in “6-DOF GraspNet: Variational Grasp Generation for Object Manipulation” by Arsalan Mousavian, Clemens Eppner, Dieter Fox. However, in that work, the authors augment the variational sampling step w/ subsequent classification and grasp adjustment in order to get accurate grasp placement. Why was that not also necessary here?

- There was a question about where the ground truth labels came from for training the model. Were these hand-labeled? How large was the dataset / how many human demonstrations were provided?

- The baseline method, “Top-1 Point”, was not described very clearly in the main text and there is no other baseline.

- There was a concern that since the method is only evaluated for two robotic arms, it is unfair to assert the method works for larger numbers of arms.

Post-rebuttal:

We appreciate the new three-arm experiment strengthens the paper and the revisions for clarity.